# Mitochondrial Fatty Acid β-Oxidation and Resveratrol Effect in Fibroblasts from Patients with Autism Spectrum Disorder

**DOI:** 10.3390/jpm11060510

**Published:** 2021-06-04

**Authors:** Rita Barone, Jean Bastin, Fatima Djouadi, Indrapal Singh, Mohammad Azharul Karim, Amrit Ammanamanchi, Patrick John McCarty, Leanna Delhey, Rose Shannon, Antonino Casabona, Renata Rizzo, Richard Eugene Frye

**Affiliations:** 1Department of Clinical and Experimental Medicine, Child Neuropsychiatry Section, University of Catania, 95124 Catania, Italy; rerizzo@unict.it; 2CNR-Institute for Polymers, Composites and Biomaterials IPCB, 95124 Catania, Italy; 3Centre de Recherche des Cordeliers, INSERM U1138, Sorbonne Université, Université de Paris, 75006 Paris, France; jean.bastin@inserm.fr (J.B.); fatima.djouadi@inserm.fr (F.D.); 4Barrow Neurological Institute at Phoenix Children’s Hospital, Phoenix, AZ 85016, USA; isingh@arizona.edu (I.S.); karim2@arizona.edu (M.A.K.); aammanamanchi@arizona.edu (A.A.); pmccarty@phoenixchildrens.com (P.J.M.); 5Department of Child Health, University of Arizona College of Medicine—Phoenix, Phoenix, AZ 85004, USA; 6Arkansas Children’s Research Institute, Little Rock, AR 72758, USA; LMDelhey@uams.edu (L.D.); SROSE@uams.edu (R.S.); 7Department of Biomedical and Biotechnological Sciences, Physiology Section, University of Catania, 95124 Catania, Italy; casabona@unict.it

**Keywords:** autism spectrum disorder, energy metabolism, fatty acid oxidation, acyl-carnitines, resveratrol

## Abstract

Patients with autism spectrum disorder (ASD) may have an increase in blood acyl-carnitine (AC) concentrations indicating a mitochondrial fatty acid β-oxidation (mtFAO) impairment. However, there are no data on systematic mtFAO analyses in ASD. We analyzed tritiated palmitate oxidation rates in fibroblasts from patients with ASD before and after resveratrol (RSV) treatment, according to methods used for the diagnosis of congenital defects in mtFAO. ASD participants (*N* = 10, 60%; male; mean age (SD) 7.4 (3.2) years) were divided in two age-equivalent groups based on the presence (*N* = 5) or absence (*N* = 5) of elevated blood AC levels. In addition, electron transport chain (ETC) activity in fibroblasts and muscle biopsies and clinical characteristics were compared between the ASD groups. Baseline fibroblast mtFAO was not significantly different in patients in comparison with control values. However, ASD patients with elevated AC exhibited significantly decreased mtFAO rates, muscle ETC complex II activity, and fibroblast ETC Complex II/III activity (*p* < 0.05), compared with patients without an AC signature. RSV significantly increased the mtFAO activity in all study groups (*p* = 0.001). The highest mtFAO changes in response to RSV were observed in fibroblasts from patients with more severe symptoms on the Social Responsiveness Scale total (*p* = 0.001) and Awareness, Cognition, Communication and Motivation subscales (all *p* < 0.01). These findings suggested recognition of an ASD patient subset characterized by an impaired mtFAO flux associated with abnormal blood AC. The study elucidated that RSV significantly increased fibroblast mtFAO irrespective of plasma AC status, and the highest changes to RSV effects on mtFAO were observed in the more severely affected patients.

## 1. Introduction

Autism spectrum disorder (ASD) is a neurodevelopmental condition characterized by early social communication deficits and repetitive motor behavior, sensory abnormalities, and restricted interests, with a global prevalence of about 1% [1] and, according to the most recent estimates, of about 2% in the United States (US) [2]. ASD biology is particularly complex, including individual genetic contributions interacting with multiple environmental factors. Genetic risk points to a complex inheritance, with additive contributions from common variants or through rare variants with larger effect sizes [3]. The vast majority of ASD is idiopathic, with a specific cause identified in only 4–20% of patients, including a genetic etiology. Association of ASD behavioral phenotypes to specific genetic subtypes is envisaged; however, patients with molecularly defined ASD are not easily clinically identified because clinical and neurobehavioral correlates of a given genetic contribution vary widely [3,4].

Immune dysregulation/inflammation, oxidative stress, and mitochondrial dysfunction are all key pathologic underpinning of ASD [5,6]. Complex biological changes play a role in ASD clinical heterogeneity, hindering the discovery of universal biomarkers for diagnosis and treatments. Currently, ASD diagnosis is based on measurements of behavioral symptoms according to the Diagnostic Statistical Manual of Mental Disorders Version 5 (DSM-5) diagnostic criteria [7]. Recent data point to identifying patient subsets to better define the contribution of certain biological changes in individual patients. In this context, considerable evidence highlights energy metabolism abnormalities in ASD, pointing to acquired mitochondrial dysfunction in a proportion of patients [8,9]. Impaired mitochondrial metabolism may influence neuronal development and synaptic plasticity, which play a major role in neurodevelopment and contribute to ASD. Low free carnitine [10] and abnormal levels of blood acyl-carnitines (AC) were repeatedly found in ASD clinical studies [11,12]. We recently tested the metabolic profile in dried blood spots to support early recognition of young children at risk for ASD diagnosis. We found a significant increase in blood short-chain and long-chain AC and, to a lesser extent, medium-chain AC. Using machine learning analyses, we found a high classification performance of this AC signature to support diagnosis at younger ages (<5 years) [13]. Interestingly, a similar pattern of increased AC had been detected in patients with ASD from the US [12] and in rodent models, where ASD-like behavior was induced by propionic acid [14]. Recently, a global metabolome analysis of plasma and feces found differential levels of AC among the most discriminant metabolites in ASD compared with typically developing (TD) populations. Furthermore, blood C2–C14 AC levels were positively correlated with a more severe impairment of social behavior, supporting a key role for mitochondrial dysfunction in ASD pathophysiology [15].

The mitochondrial fatty acid β-oxidation (mtFAO) is a major energy-producing metabolic pathway, which uses fatty acids to produce adenosine triphosphate (ATP). First, the import of long-chain fatty-acids (LCFA) into mitochondria requires the activity of a multi-enzymatic carnitine-dependent shuttle, with formation of AC intermediates. Within mitochondria, AC are converted back to acyl-CoA, and the various β-oxidation enzymes isoforms progressively shorten the acyl-CoA to produce acetyl-CoA, NADH, and FADH_2_ via the Lynen helix [16]. Re-oxidation of produced NADH and FADH_2_ by the mitochondrial respiratory chain ultimately results in the production of large amounts of ATP. In addition, very recent data unveiled the involvement of mtFAO in instructing non-energy-related functions, such as chromatin modification or neural stem cell activity [17,18].

Thus, AC enables the transport of fatty acids across mitochondrial membranes, and these conjugated fatty acids typically accumulate when β-oxidation is disturbed. This led to widespread clinical applications in the screening for inborn mtFAO disorders in newborns, implemented in many countries, which is based on the detection of specific accumulating AC in the blood of newborns [19]. In line with this, it can be hypothesized that the increase in plasma AC levels in patients with ASD points to an impairment of mtFAO. To date, the implication of mtFAO in the pathophysiology of ASD is largely unknown, but there are consistent findings documenting AC accumulation, and thereby possible mtFAO impairments, as biomarkers and therapeutic targets in a subset of patients with ASD. In the present study, we therefore analyzed the mtFAO flux and respiratory chain activities on primary cultured fibroblasts from ASD patients with elevated plasma AC (w-AC) or without elevated plasma AC (w/o-AC).

Resveratrol (RSV) (3,5,4-trihydroxy-trans-stilbene) is a natural polyphenol produced in plants and enriched in grapes and red fruits. Pre-clinical studies showed that RSV ameliorated social behavior and sensory alterations in the rat model of ASD induced by valproic acid [20,21]. At the molecular level, RSV decreased neuroimmune dysregulation through the inhibition of neuronal toll-like receptors and COX-2 signaling [22] and by downregulation of the chemokine receptor in the BTBR T^+^ Itpr3^tf^/J mice model [23]. Interestingly, RSV was also shown to stimulate mtFAO in control human fibroblasts and could restore normal mtFAO rates in fibroblasts from patients with mild forms of inborn mtFAO deficiencies [24]. This further supports a putative therapeutic effect of RSV in patients with ASD [25] and led us to test whether RSV might induce up-regulation of mtFAO in fibroblasts from ASD patients.

Altogether, the present study had several objectives. At first, we aimed to determine if fibroblasts from ASD patients exhibited mitochondrial mtFAO and respiratory chain deficiencies, compared with fibroblasts from control individuals. In parallel, we sought to determine whether cellular mtFAO rates correlated with the AC status of the ASD patients. Then, we tested the effects of RSV on fibroblasts mtFAO rates in patients with ASD and we evaluated whether the mtFAO response upon RSV treatment was depended on clinical characteristics.

## 2. Materials and Methods

### 2.1. Participants

Protocols used in this study were registered in clinicaltrials.gov as NCT02000284 and NCT02003170 and approved by the Institutional Review Board at the University of Arkansas for Medical Sciences (Little Rock, AR, USA). Parents of participants provided written informed consent. All participants were recruited from the Arkansas Children’s Hospital Autism Multispecialty clinic directed by Dr Richard E. Frye (senior author). The ASD diagnosis was documented by at least one of the following: (i) a gold-standard diagnostic instrument, such as the Autism Diagnostic Observation Schedule and/or Autism Diagnostic Interview-Revised (ADI-R); (ii) the state of Arkansas diagnostic standard, defined as agreement of a physician, psychologist, and speech therapist who specializes in ASD; and/or (iii) Diagnostic Statistical Manual of Mental Disorders diagnosis by a physician along with standardized validated questionnaires including the Social Responsiveness Scale (SRS), the Social Communication Questionnaire and the Autism Symptoms Questionnaire, all of which have excellent correspondence to the gold-standard instruments, along with diagnosis confirmation by the referral investigator (senior author). In our recent clinical trial [26], we found that methods (ii) and (iii) were consistent with the ADI-R diagnostic criteria for ASD.

In general, fibroblast samples were obtained for clinical use and then transferred to the research laboratory. For individuals that underwent sedated procedures, most commonly muscle biopsy, the samples were obtained under sedation by the surgeon. For individuals that did not undergo other procedures, Dr Richard E. Frye (senior author) personally obtained the sample by a punch biopsy with local anesthesia.

Overall, fibroblast samples were available from 10 children diagnosed with ASD for this study, 6 males and 4 females aged 7.4 ± 3.2 years (mean ± SD) (range: 3–13). Seven patients (70%) were diagnosed with regressive ASD. Five control fibroblasts from children of similar age who did not manifest any known medical disease or genetic abnormalities were obtained from Coriell Institute for Medical Research (Camden, NJ, USA).

### 2.2. Neurodevelopmental and Behavioral Measurements

Neurodevelopment assessment was accomplished by the Vineland Adaptive Behavior Scale (VABS) 2nd edition. The VABS is a valid tool based on structured interview with a caretaker allowing to measure age-appropriate abilities in everyday skills including social and motor skills, communication, and daily living. The VABS provides standard scores (m = 100, SD = 15) and higher scores indicate better functioning. The Aberrant Behavior Checklist (ABC), a 58-item parent-reported questionnaire, was used to measure behavioral symptoms across 5 subscales (social withdrawal, hyperactivity, stereotypy, inappropriate speech, and irritability) (0–30 raw scores, higher is worse). Multiple ASD clinical trials have used it and it has both convergent and divergent validity [26]. ASD symptoms severity was assessed by the SRS, a 65-item questionnaire completed by a parent or close family member that measures the severity of social skill deficits across five domains (awareness, cognition, communication, motivation and restricted interests, and repetitive behavior) (clinical cut-off ≥ 60).

### 2.3. Cell Culture and Metabolic Evaluation

Fibroblasts were derived from skin biopsies obtained with written informed consent from the parents. Control and patient fibroblasts were cultured at 37 °C, 5% CO_2_ in RPMI with Glutamax™ (Carlsbad, CA, USA) supplemented with 10% (*v*/*v*) fetal bovine serum and 0.2% (*v*/*v*) primocin (InvivoGen, San Diego, CA, USA). For treatment, the media were removed and vehicle (0.04% DMSO) or RSV (75 µM *trans*-RSV, Cayman Chemical, Bertin technologies, Montigny-le-Bretonneux, France), were added to a fresh medium for the last 48 h of culture before mtFAO measurement, as previously described [27]. The mtFAO flux was measured in cultured fibroblasts from w-AC (*N* = 5) and w/o-AC (*N* = 5) patients, and in control fibroblasts (*N* = 5) obtained from healthy subjects with equivalent age and gender distribution. Metabolic evaluation was blinded to patient group, specifically fibroblast assignments were retained confidential and were unveiled at the end of the mtFAO assay for statistical analysis. The FAO flux was determined by quantifying the production of ^3^H_2_O from (9,10-^3^H) palmitate, as described previously [27]. The FAO assay was run in triplicate and was repeated twice for each cell line. The FAO assay on resveratrol-treated fibroblasts was performed once. Results (mean ± SD) were expressed in nmol of tritiated-palmitate oxidized per hour per milligram of protein. The electron transport chain (ETC) activity was tested in frozen muscle biopsies and/or cultured fibroblast cultures as previously reported (Baylor Medical Genetics Laboratory, Houston TX, USA) [28]. Values corrected and uncorrected for citrate synthase (CS) activity were considered.

### 2.4. Statistical Analyses

Data were presented as means and standard deviations (SDs) for continuous variables. Statistical analyses on FAO rates were conducted in the three groups of fibroblasts (w-AC, w/o-AC, and controls). The data were preliminarily subjected to the Shapiro–Wilk test to verify the presence of a normal distribution of the sample, and to the Levene’s test to verify the homogeneity of the variances between the groups. Two-way analysis of variance (ANOVA) with repeated measures was used to verify whether FAO was influenced by the group itself (inter-group effect) and presence/absence of RSV (intra-group effect). We used the ANOVA to compare the three groups and all combinations of paired groups. Possible differences between paired groups in the ANOVA analyses were then evaluated by *t*-test with the Bonferroni correction. Student’s *t*-test was used to compare the mean values of ETC activity and the clinical scores between the ASD study groups. Correlations among study variables were analyzed by the Pearson correlation analysis. Differences with *p* < 0.05 were considered significant. We assumed that the data were normally distributed based on previous studies which have examined these measures in larger sample sizes as well as by examination of our current dataset. Thus, parameter statistical analyses—especially the technique used here within which are robust to small differences in parameter distribution—were considered appropriate for analysis.

Additionally, to interpret differences between groups, especially in the context of small sample sizes where biological variability can prevent differences from being statistically significant, we compared differences between groups with the minimally clinical important difference (MCID), a value which indicates whether the difference could be considered clinically significant. These values for the VABS, SRS, and ABC are provided in previous clinical studies [26]. Data were analyzed using the SPSS Statistics software, version 23 (SPSS, Inc., Chicago, IL, USA, IBM, Somers, NY, USA).

## 3. Results

### 3.1. Blood AC Levels in Patients with ASD

Participants were divided in two groups: w-AC (n:5; age 7.2 ± 3.7) or w/o-AC (n:5; age 7.6 ± 3.1). Mean age was not significantly different in the two study groups (*p* = 0.429). Patients with high plasma AC levels had consistent AC elevation defined as at least three AC significantly elevated (*p* < 0.05) in repeated analyses [12] (Figure 1).

### 3.2. mtFAO Activity in Fibroblast Cultures from Patients with ASD

There was no overall significant difference in FAO rates between controls and the ASD fibroblasts (Figure 2; F_1,8_ = 1.107; *p* = 0.368) indicating that under basal conditions, FAO was not significantly impaired in ASD children’s fibroblasts, regardless of the AC status. However, significant differences were found between the two ASD patient groups (F_1,8_ = 5.374; *p* = 0.049). Pairwise *t*-test revealed that the effect found between the ASD groups in the ANOVA was driven by differences between the two groups of children with ASD, before RSV supplementation. In fact, under basal conditions, the w-AC patients’ fibroblasts exhibited significantly decreased FAO values (5.20 ± 0.42) compared with those measured in the w/o-AC group (5.9 ± 0.53) (*p* = 0.044). Basal FAO rates (5.85 ± 1.12) appeared higher in the control compared with the w-AC patients, but no significant differences were found in paired comparisons, likely because of higher variability of FAO rates in controls. Overall, although not significantly impaired compared with controls, baseline FAO values were significantly lower in patients w-AC elevations compared with patients with normal AC blood levels (Figure 2).

### 3.3. Electron Transport Chain Complex Activity

Measurements of activities of the ETC complexes were performed on frozen muscle biopsies (*N* = 8) and cultured fibroblasts (*N* = 10). Percentages of normal ETC function, uncorrected or corrected for citrate synthase, were compared between the w-AC and w/o-AC groups (Figure 3).

The graph values represent percentages of normal ETC function, uncorrected (Figure 3A,C) or corrected for citrate synthase (Figure 3B,D). Activities of the complex II in the muscle (M; Figure 3B) and the complex II/III in fibroblasts (FB; Figure 3D) were significantly reduced in the patients with ASD with plasma AC elevations as compared with patients with ASD without plasma AC elevations (*p* < 0.05).

### 3.4. Clinical Characteristics

Standardized clinical assessment in participants with ASD is reported in Appendix A. On average, VABS indicated mildly impaired functioning in all three domains [daily living SS (70.1 ± 16.3), communication SS (73.1 ±16.3); and socialization SS (70.5 ± 16)] in all participants with ASD. The levels of functional disability for ASD symptoms, communication, and socialization measured by the VABS subscales were lower in the participants without AC elevations as compared with those with AC elevations, and these differences exceeded the MCID, suggesting that they were clinically observable. However, these differences were not statistically significant, probably because of the small sample size. On average, patients had disruptive behavioral symptoms, such as irritability and hyperactivity, at least of moderate severity (defined as scores greater than or equal to 13 on the ABC). Patients in the w-AC group had higher scores on the ABC social withdrawal, inappropriate speech, and stereotypy subscales compared with the w/o-AC group, and these differences exceeded the MCID, suggesting that they were clinically observable. Patients in the w/o-AC group had higher scores on the ABC hyperactivity and irritability, and they exceeded the MCID. However, all these differences were not statistically significant, probably because of the small sample size.

All participants with ASD had social impairment severity in the clinical range (T-scores > 60 on SRS total), with no significant differences between the study groups. However, those with ASD with AC elevations demonstrated less impaired social awareness and cognition but greater repetitive behaviors (mannerism) on the SRS as compared with those with ASD but without AC elevation, and these differences exceeded the MCID. 

Then we evaluated whether fibroblast mtFAO at baseline and upon RSV treatment was depended on clinical characteristics (Appendix A). We found that patients who had the highest mtFAO activity in response to RSV were the most impaired on the SRS–total (*r* = 0.65; *p* = 0.044), SRS–awareness (*r* = 0.72; *p* = 0.019), and SRS–cognition (*r* = 0.76; *p* = 0.011). Moreover, the highest changes to RSV with respect to baseline were observed in patients most impaired on the SRS–total (*r* = 0.87; *p* = 0.001; Figure 4A) as well as on SRS subscales including awareness (*r* = 0.89; *p* < 0.001; Figure 4B), cognition (*r* = 0.88; *p* < 0.001; Figure 4C), communication (*r* = 0.78; *p* = 0.008; Figure 4D), and motivation (*r* = 0.77; *p* = 0.009; Figure 4E).

## 4. Discussion

In the present study, we systematically applied palmitate oxidation rate measurements in fibroblasts to investigate ex vivo mtFAO levels in a series of patients with ASD. We found that under basal conditions, mtFAO in fibroblasts was not significantly different in children with ASD compared with healthy controls. However, the mtFAO flux was significantly decreased in patients with elevated blood AC compared with patients with normal blood AC. Significant differences in basal mtFAO rate between ASD groups may be consistent with recurrent findings of elevated blood AC in subgroups of children with ASD [12,13].

As mentioned above, screening of inborn mtFAO deficiencies in newborns is based on acyl-carnitines analysis in dried blood spots, which needs to be followed by confirmatory testing to define the ultimate diagnosis [19]. Importantly, patients diagnosed in this way at birth are mostly asymptomatic, and the AC pattern is not predictive of symptomatology [29]. In many cases, the symptom-free period will last for years, and the disease manifestations will only develop in the adolescence or adulthood, generally triggered by exercise, cold, fever, or other metabolic stress situations. This is true in particular for the VLCAD deficiency, one of the most common inborn mtFAO disorders. In the mild—most frequent—form of this disorder, the typical C (14:1) AC accumulation detected at birth can be associated with near-normal FAO flux values in the patients’ fibroblasts and with the absence of clinical manifestations [30,31,32]. In adult patients with the muscular form of carnitine-palmitoyl transferase II (CPT-II) deficiency, high plasma levels of long-chain AC can be measured in the absence of disease symptoms, and mtFAO values in patient fibroblasts may also appear marginally modified [33]. In sum, evidence from genetic mtFAO defects suggests that an inefficient mtFAO flux with accumulation of AC biomarkers may arise without obvious symptoms affecting mtFAO-dependent organs and even in the presence of a relatively high residual mtFAO activity. The occurrence of developmental delay, autistic-like behavior, or ASD in genetic defects of mtFAO, particularly VLCAD [34] and LCHAD [35], suggests that impaired mtFAO may contribute to dysfunctional energy metabolism in subsets of patients with ASD. Furthermore, interestingly, post-hoc analysis of newborn screening data in a large (>9000 individuals) cohort showed that high levels of ACs could be associated with an increased risk of ASD [36]. Specific abnormalities in AC also characterize an animal model of ASD in which propionic acid, a microbiome-produced short chain fatty acid, results in ASD-like behavior, mitochondrial dysfunction, and neuroinflammation [37].

In addition to reduced mtFAO oxidation rate, patients with ASD and plasma AC elevations had significantly lower ETC complex II and complex II/III activities in muscle and fibroblasts, respectively, when compared with patients with normal AC blood levels. This is consistent with a previous study that demonstrated a partial deficit in complexes I/III and I/III RS functions in muscle of patients with ASD and elevated blood AC [12]. As a whole, the current findings support recognition of an ASD patient subset characterized by impaired mtFAO efficiency associated with abnormal blood AC underlying decreased energy production in different cell types.

It is noteworthy that our finding of a >200% increase in the ETC complex I/III activity in patients with ASD is consistent with recent findings illustrating that an elevation of the respiratory chain activity may be abnormal and associated with ASD. An increase in the baseline respiratory chain activity in lymphoblastoid cell lines from subsets of patients with ASD is associated with increased vulnerability to environmental toxicants and to physiological stress [38,39]. It was shown that patients with ASD and mitochondrial dysfunction have higher rates of neurodevelopmental regression with loss of acquired abilities following fever or infections [40] as not uncommonly seen in patients with ETC mitochondrial disorders [41]. Such abnormalities are consistent with acquired mitochondrial dysfunction in ASD patients with neurodevelopmental regression [42] and with studies which demonstrated that mitochondrial dysfunction in ASD can be associated with environmental factors, such as fever [40], immune activation and oxidative stress [43], microbiome metabolites [37], air pollution [44], and prenatal nutritional deficiencies [45]. Some other symptoms of mitochondrial disease, including hypotonia and motor delay, seizures, and gastrointestinal disturbances are frequently encountered in ASD as well [5].

Previous research established that ex vivo measurements of palmitate oxidation rate provide a sensitive approach to test the efficiency of compounds potentially capable to stimulate mtFAO, such as fibrates and RSV [24,27,46]. Pre-clinical studies using different ASD models showed the effects of RSV on several pathways involved in ASD, such as decreasing microglia-induced neuroinflammation and oxidative stress, by reducing oxygen and nitrogen reactive species and neurotransmitter imbalance [22,47]. More recently, RSV was found capable of ameliorating social behavioral deficits in the oxytocin receptor gene knockout by up-regulation of the silent information regulator 1 (Sirt1) gene and early growth response factor 3 (Egr3) gene expressions in the amygdala of Oxtr-KO mice [48]. In 2020, a double-blind, placebo-controlled clinical trial investigated the effects of RSV as an adjunctive treatment in decreasing disruptive behavioral symptoms measured by SRS in patients with ASD. RSV add-on therapy to Risperidone did not yield any significant improvement on the irritability subscale compared with placebo but led to significant improvement on the hyperactivity/non-compliance subscale after a 10-week period [49]. No severe adverse effects were observed and no significant difference in the frequency of adverse effects was documented between RSV-treated versus placebo group, thus supporting further clinical studies using RSV monotherapy in patients with ASD.

For the first time, the current study explored the effects of RSV on mtFAO in fibroblasts from patients with ASD. It showed that RSV significantly increased mtFAO activity in fibroblasts from patients with ASD as well as in controls. RSV effects on FAO are mediated by peroxisome proliferator activated receptor gamma co-activator-1-alpha (PGC-1α). PGC-1α serves as a co-activator of various transcription factors [PPARs, NRF 1 and 2 (nuclear respiratory factors), hepatic nuclear factor 4], thus regulating the expression of several enzymes involved in the mtFAO pathway, mitochondrial biogenesis, oxidative phosphorylation, and energy production [24]. Importantly, we showed an association between RSV effects on mtFAO activity in fibroblasts from children with ASD and pertinent clinical characteristics. We found that the highest changes to RSV with respect to baseline were observed in patients with more severe ASD symptoms on the SRS scale, such as impaired awareness, cognition, social communication, and motivation.

The main limitations of the present study include the small size of the samples used for ex vivo analyses of mtFAO. This suggests that the study is underpowered, so non-significant differences cannot be claimed to signify that no difference exists. Clearly, further studies with larger sample sizes will be needed to follow up this work. Secondly, it should be noted that results from pre-clinical studies such as the present one may partially apply to the in vivo condition and will ultimately require to be tested in clinical trials. As ASD is heterogeneous in nature, there is a need to characterize patients also considering what possible differences in metabolic profiling might lead to different clinical responses to the same therapies. Taken into account that ASD is unequivocally associated with mtFAO impairment in patient subsets, the effects of RSV in ameliorating mtFAO in patient cells may be relevant for understanding the therapeutic response.

## 5. Conclusions

In conclusion, we found a significant difference in fibroblasts basal mtFAO rates between ASD groups that differ by the presence or absence of elevated blood acyl-carnitines. The study showed for the first time that the mtFAO activity in fibroblasts of ASD children increased significantly after RSV and that the highest changes to RSV effects on mtFAO occurred in the most severely affected patients. In the light of the present findings, future clinical trials might well consider whether possible RSV effects in ameliorating behavioral symptoms are associated with baseline symptoms severity in patients with ASD. Moreover, further studies should take into account how and to what extent different metabolic profiles may influence response to therapies in patients with ASD.

## Figures and Tables

**Figure 1 jpm-11-00510-f001:**
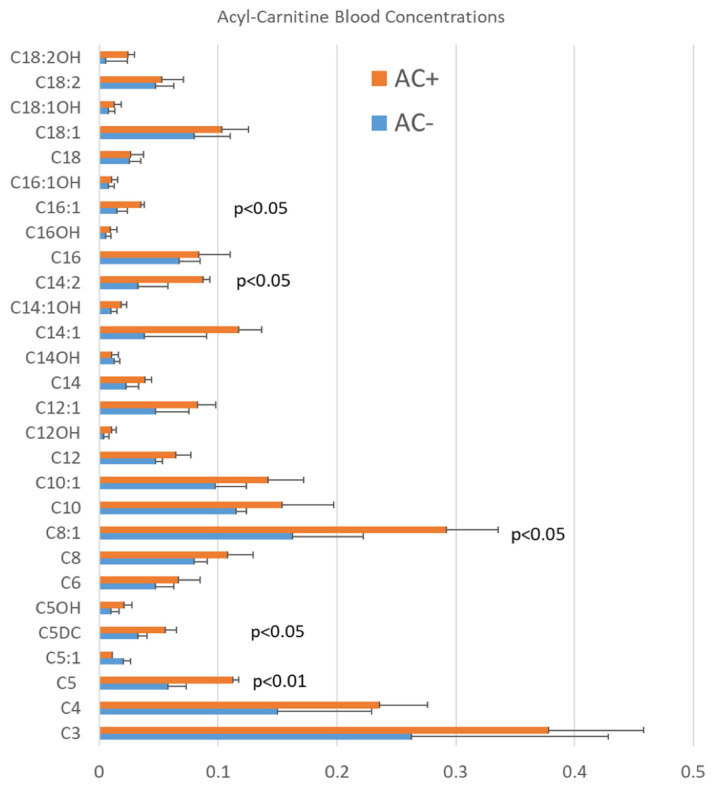
Average blood acyl-carnitine (AC) levels in studied patients with autism spectrum disorder (ASD) with (AC+, orange bars) and without (AC−, blue bars) elevations in acyl-carnitines. Short-chain (C5, C5DC), medium-chain (C8:1), and long-chain (C14:2, C16:1) acyl-carnitines (AC) were significantly elevated in a subset of studied patients with ASD. Patients with high plasma AC levels had consistent AC elevation defined as at least three AC clinically significantly elevated (outside the normal reference range) in repeated analyses.

**Figure 2 jpm-11-00510-f002:**
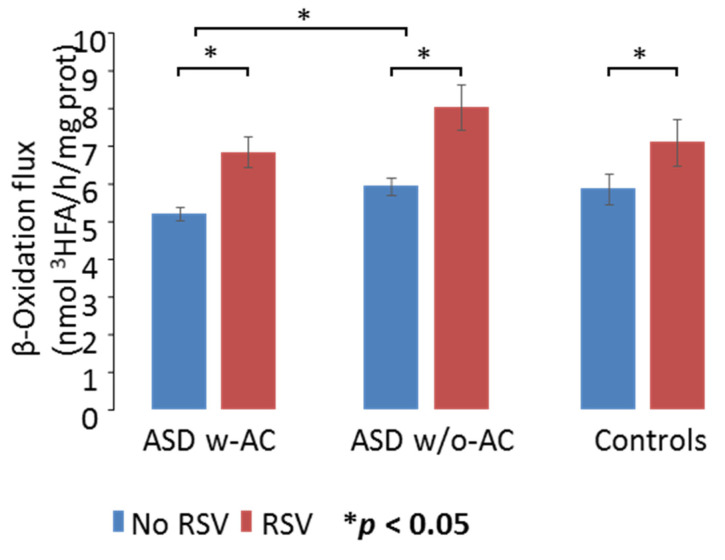
Mitochondrial fatty acid oxidation (FAO) rates (blue bars) and effects of resveratrol (RSV) (red bars) in fibroblasts of ASD patients and healthy control individuals. Figure shows participants with autism spectrum disorder (ASD) that also have acyl-carnitine (AC) elevations (ASD w-AC) and participants with ASD without AC elevations (ASD w/o-AC) as well as control participants. Under basal conditions, the β-oxidation flux (nmol ^3^HFA/h/mg protein) measured in fibroblasts of patients with ASD w-AC were significantly decreased compared with the ASD w/o-AC. RSV significantly increased mtFAO values in all the study groups. The results are means (±SD) of three different experiments. * *p* < 0.05.

**Figure 3 jpm-11-00510-f003:**
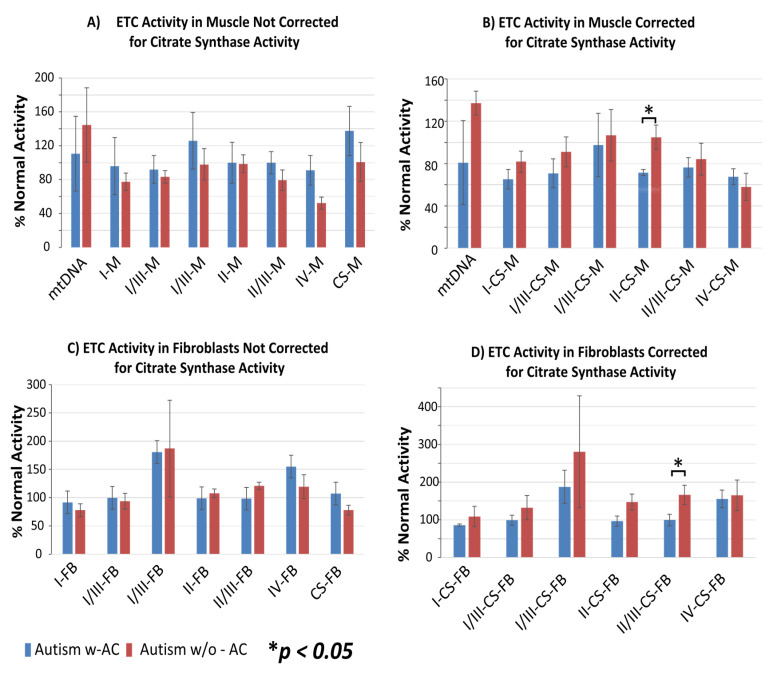
Electron transport chain (ETC) activity in muscle (M; Panels (**A**,**B**)) and fibroblast (FB) culture (Panels (**C**,**D**)) without (Panels (**A**,**C**)) and with (Panels (**B**,**D**)) correction for citrate synthase (CS) activity, in patients with autism spectrum disorder (ASD) that also have acyl-carnitine (AC) elevations (ASD w-AC, blue bars) and participants with ASD without AC elevations (ASD w/o-AC, red bars). Mitochondrial DNA copy number (mtDNA) is also provided in the graphs for muscle samples.

**Figure 4 jpm-11-00510-f004:**
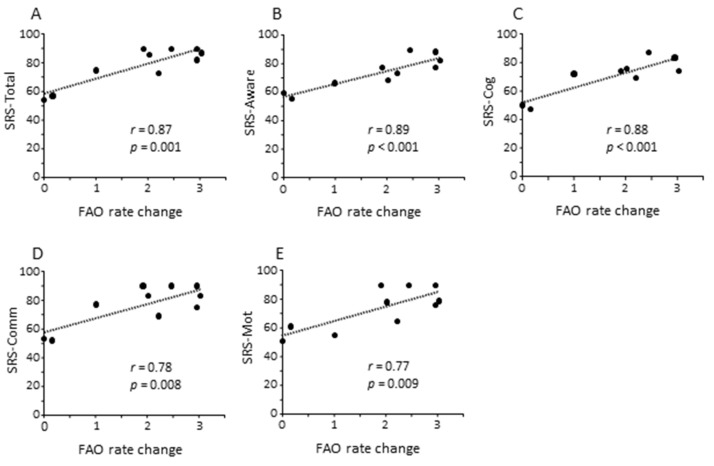
Correlation analyses among clinical characteristics of patients with ASD as measured by the Social Responsiveness Scale (SRS) and resveratrol (RSV) effect on fatty acid oxidation (FAO) in fibroblasts. Highest changes of mtFAO in response to RSV were significantly associated with clinical impairment (higher scores) on the SRS–total (**A**) as well as on SRS awareness (**B**), cognition (**C**), communication (**D**), and motivation (**E**) subscales.

## Data Availability

All data presented in the study are included in the article. Further inquiries can be directed to the corresponding authors.

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
