# Peer review of "Mitochondrial Fatty Acid β-Oxidation and Resveratrol Effect in Fibroblasts from Patients with Autism Spectrum Disorder"

_jpm, 2021, doi:10.3390/jpm11060510_

Round 1

Reviewer 1 Report

Barone et all explored the effects of RSV on mitochondrial FAO in fibroblasts of patients with ASD. They evaluated the correlation between mtFAO rates and Acylcarnitine levels, the clinical characteristics and the effect of RSV on these aspects.

Overall, this is an interesting subject and the study is well-planned and executed.

Ich have only small remarks:

The abbreviations and axis titles have to be defined in legends of figures 2, 3 and 4.

In Figure 3, the authors have to define the different panels (a-d). It should be explained detailed in the figure about the significance state. For example, regarding the data shown (in panel b maybe) I cannot imagine, that there is any significant difference between KI/III and II-CS in w/o-groups. It could be instead between w and w/o groups, which has to be defined exactly.

Author Response

Point-by-point letter (JPM-1228025)

Reviewer #1

Comment: Overall, this is an interesting subject and the study is well-planned and executed.

Response: We thank the reviewer for the positive comment and remarks.

Comment: The abbreviations and axis titles have to be defined in legends of figures 2, 3 and 4.

Response: abbreviations and axis titles have now been clarified in Figures 1,2,3,4

In Figure 3, the authors have to define the different panels (a-d). It should be explained detailed in the figure about the significance state. For example, regarding the data shown (in panel b maybe) I cannot imagine, that there is any significant difference between KI/III and II-CS in w/o-groups. It could be instead between w and w/o groups, which has to be defined exactly.

Response: The panels have now been defined. The significant is outlined by a bar and a star. The difference in I/III-CS are shown in panel D. This has now been clarified in the text.

Reviewer 2 Report

In the present work mitochondrial fatty acid β-oxidation and resveratrol effect in fibroblasts from patients with autism spectrum disorder has been studied. I believe the work is of interest for a reader of Journal of Personalized Medicine. 
The manuscript is generally well written. However, I have some concerns regarding the sample size used and the power calculation along with stats used. The work and statistics appears as kind of neglected or scarce or naive and there are some aspects that are not very clear to me. I believe that the paper may deserve publication but it should be revised. 

  1. Sample size and power calculation: The specific calculation of the simple size is not expressed and the subjects evaluated are too few to be able to draw any conclusion. Please, provide the formula used by the authors for power calculation.
  2. There are a number of possible weak points for the use of a full parametric approach here. In the statistical methods the authors consider variables with a normal distribution. It is not reported how the fitting to normality was assessed. This should be reported even if it was a vision inspection of the histogram plot. The authors should consider that normality of the marginal distribution (by comparison group) is the prerequisite for comparisons by groups using means. I believe that assumption of normality here, considering also small sample sizes, is very difficult to be assumed. I would have performed a fully non parametric approach, at least as a supplementary sensitivity analysis. Please also consider that other numerous prerequisite for those statistical analysis should be checked such as homogeneity of variances. The authors could have considered to apply a data transformation.
  3. Furthermore, the authors not used an “adjusted” approach. Why authors not considered confounder variables for analysis? I believe that it is very important. Please provide an adjusted analysis and mostly important why those factors were considered and how were selected.
  4. Please, explain tables footer. Information is scarce. In table S1 two groups were compared?
  5. Regarding the ABC test, additional information is needed to help the reader contextualize the results. What are meaningful changes/ranges for the subscales? It’s hard to tell whether some of the results that are statistically significant are clinically meaningful for readers not familiar with this instrument.

Author Response

Point-by-point letter (JPM-1228025)

Reviewer #2

In the present work mitochondrial fatty acid β-oxidation and resveratrol effect in fibroblasts from patients with autism spectrum disorder has been studied. I believe the work is of interest for a reader of Journal of Personalized Medicine.

The manuscript is generally well written. However, I have some concerns regarding the sample size used and the power calculation along with stats used. The work and statistics appears as kind of neglected or scarce or naive and there are some aspects that are not very clear to me. I believe that the paper may deserve publication but it should be revised.

We thank the Reviewer for the positive comment and remarks.

Comment: Sample size and power calculation: The specific calculation of the simple size is not expressed and the subjects evaluated are too few to be able to draw any conclusion. Please, provide the formula used by the authors for power calculation.

Response: The reviewer is correct that the sample size is small so the study is underpowered and we have not stated this in the limitations. However, power calculations are typically used in clinical trials in order to informing whether non-significant statistical values can be claimed to signify that there are no difference between the groups compared. We are not claiming that any non-significant findings indicate no significant difference, we are concentrating on the positive findings.

Following this Reviewer’s remark, we highlight among study limitations that :”The main limitations of the present study include the small size of the samples used for ex-vivo analyses of mtFAO. This suggest that the study is underpowered so non-significant differences cannot be claimed to signify that no difference exists. Clearly further studies with larger sample sizes will be needed to follow-up this work.”

Comment: There are a number of possible weak points for the use of a full parametric approach here. In the statistical methods the authors consider variables with a normal distribution. It is not reported how the fitting to normality was assessed. This should be reported even if it was a vision inspection of the histogram plot. The authors should consider that normality of the marginal distribution (by comparison group) is the prerequisite for comparisons by groups using means. I believe that assumption of normality here, considering also small sample sizes, is very difficult to be assumed. I would have performed a fully non parametric approach, at least as a supplementary sensitivity analysis. Please also consider that other numerous prerequisite for those statistical analysis should be checked such as homogeneity of variances. The authors could have considered to apply a data transformation.

Response: The data were preliminarily subjected to the Shapiro-Wilk test to verify the presence of a normal distribution of the sample, and to the Levene’s test to verify the homogeneity of the variances between the groups. The above statement has been reported in the statistical analyses description. Although the Reviewer is correct that our sample is too small to rigorously investigate the shape of the data distribution, these values have been used in many other studies where normality of the distribution has been assumed and parametric statistical tests have been used. Additionally, our statistical techniques are robust to small variations in parameter distribution. We are not sure why the reviewer would like us to use data transformation as the reviewers comment already states that our sample size is too small to reliably investigate changes in the normality of the distribution so we’re not sure how the reviewer thinks we could reliably investigate different distributions reliably with our current sample size.

Comment: Furthermore, the authors not used an “adjusted” approach. Why authors not considered confounder variables for analysis? I believe that it is very important. Please provide an adjusted analysis and mostly important why those factors were considered and how were selected.

Response: The Reviewer has already pointed out that our sample size is small. Thus, adding additional variables to the analysis is not appropriate as it would (1) distort the main statistical comparisons and (2) not be accurate because of low case numbers for any confounder.

Comment: Please, explain tables footer. Information is scarce. In table S1 two groups were compared?

Response: The abbreviations in the tables have been moved up to the section titles to make the table clearer and the caption has been expanded to clarify the groups.

Comment: Regarding the ABC test, additional information is needed to help the reader contextualize the results. What are meaningful changes/ranges for the subscales? It’s hard to tell whether some of the results that are statistically significant are clinically meaningful for readers not familiar with this instrument.

Response: To assist in the interpretation of the ABC and other standardized tests, we have now listed the minimally clinically important difference in order to understand whether the differences between groups observed would be clinically significant (Table S1 and results section). Multiple ASD clinical trials have used the ABC and it has convergent and divergent validity. This information has been added to the manuscript as has a reference [26].
